# Structured 4D Latent Predictive Model for Robot Planning

**Zhiyi Li** [1]  **Peilin Wu** [* 2]  **Xiaoshen Han** [* 2]  **Ruojin Cai** [2]  **Yilun Du** [2]

## Abstract

Video predictive models are emerging as a powerful paradigm in robotics, offering a promising path toward task generalization, long-horizon planning, and flexible decision-making. However, prevailing approaches often operate on 2D video sequences, inherently lacking the 3D geometric understanding necessary for precise spatial reasoning and physical consistency. We introduce a *Structured 4D Latent Predictive Model*, which predicts the evolution of a scene's 3D structure in a structured latent space conditioned on observations and textual instructions. Our representation encodes the scene holistically and can be decoded into diverse 3D formats, enabling a more complete and 3D consistent scene understanding. This structured 4D latent predictive model serves as a planner, generating future scenes that are translated into executable actions by a goal-conditioned inverse dynamics module. Experiments demonstrate that our model generates futures with strong visual quality, substantially better 3D consistency and multi-view coherence compared to state-of-the-art video-based planners. Consequently, our full planning pipeline achieves superior performance on complex manipulation tasks, exhibits robust generalization to novel visual conditions, and proves effective on real-world robotic platforms. Our website is available at https://structured-4d-model.github.io/.

## 1. Introduction

Learning a general-purpose agent that can solve a wide range of real-world tasks has always been a central goal in robotics. However, progress is constrained by the scarcity of large-scale, task-diverse, and interactive robotic data required to train such agents. As a result, recent work has focused on policy-based agents that directly map observations to actions (Lillicrap et al., 2015; Chi et al., 2023; Zhao et al., 2023; Xiong et al., 2021). Although these end-to-end policies can perform well in narrow, well-instrumented settings, they commonly fail to generalize under even modest distribution shifts, such as changes in lighting, viewpoint, or the composition of unseen tasks.

An alternative paradigm is to learn a dynamics model that predicts the consequences of actions, enabling planning and improving generalization. Traditionally, dynamic models have been used extensively in various robotic tasks such as locomotion and manipulation, enabling effective planners like model predictive control (MPC) to operate on top of them (Garcia et al., 1989; Mayne et al., 2000; Qi et al., 2025). Recent work in robot learning revives this approach by learning video generative models from large scale datasets and combining them with planners or inverse dynamics modules (Du et al., 2023b; Yang et al., 2023). Such models predict how the environment will evolve conditioned on text or task specifications. This makes decision-making more interpretable and flexible, facilitates long-horizon planning, and improves generalization to unseen tasks and environments. However, video-based predictive models are inherently 2D and operate in pixel space, resulting in physical inconsistencies with the real 3D world and limiting accurate spatial understanding. This limitation becomes especially problematic in fine-grained manipulation tasks where accurate 3D cues are essential (Zhu et al., 2024; Ke et al., 2024).

Modeling the dynamics of a scene directly in 3D is challenging. Traditional 3D representations, such as point clouds and meshes, preserve geometry but lose rich visual detail necessary for semantic understanding. Photorealistic representations like Neural Radiance Fields (NeRFs) (Mildenhall et al., 2021) or 3D Gaussian Splatting (Kerbl et al., 2023) better capture appearance, but are computationally intensive and not easily amenable to dynamic modeling. A common compromise is to predict RGB video along with depth and normals (Zhen et al., 2025), which provide partial 3D cues but still reduce to surface-level projections, leaving them vulnerable to occlusions and viewpoint shifts. This leads to a question, *Can we build a model that inherently simulates dynamic 3D structures of the world?*

In this paper, we propose a **Structured 4D latent predic-**

---

*Equal contribution [1]MIT [2]Harvard University. Correspondence to: Zhiyi Li <zhiyi24@mit.edu>.

*Proceedings of the 43$^{rd}$ International Conference on Machine Learning*, Seoul, South Korea. PMLR 306, 2026. Copyright 2026 by the author(s).

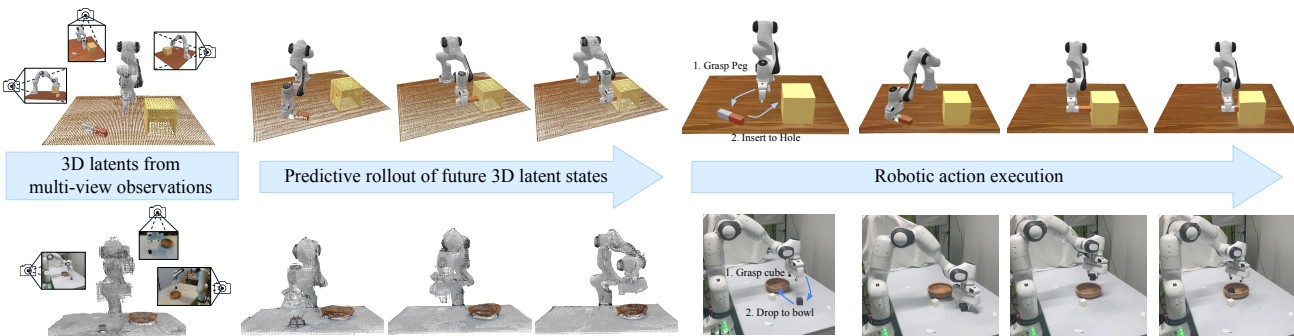

*Figure 1.* Our structured 4D latent predictive model integrates multi-view images and text instructions to forecast future 3D dynamics for robot planning and execution, demonstrated in simulation (top) and on a real robot (bottom).

**tive model** for robot planning (Figure 1). Following the success of Latent Diffusion Models (Rombach et al., 2022) which utilize spatially-aware 2D feature maps rather than unstructured 1D global latents, we adopt a structured 3D latent representation (Xiang et al., 2025) for 3D scenes. Specifically, we encode the scene into a sequence of sparse voxel grids where each active voxel holds a compact feature vector. The *grid latent* design allows us to maintain explicit 3D spatial biases, while benefiting from the computational efficiency and semantic abstraction of a low-dimensional latent space. Based on the structured 3D latents, our model learns the dynamics for the 3D scene and generates plausible future latents conditioned on current observations and text instructions. Unlike methods restricted to surface maps or videos, our latent captures holistic 3D information of the scene that can be decoded into various formats, such as 3D Gaussian Splatting. This approach enables our predictive model to achieve a more complete understanding of 3D structures and generate futures with superior 3D consistency. This detailed 3D information is then leveraged by a goal-conditioned inverse dynamics module, which translates the generated futures into precise robot actions and is especially effective for fine-grained, 3D-aware tasks. In summary, our contributions are as follows:

i) We introduce a structured 4D latent predictive model that predicts future 3D structures conditioned on current observations and textual goal, achieving comparable visual quality, strong 3D consistency, and robust viewpoint generalization.

ii) We propose a planning framework that leverages our model's detailed 3D predictions as geometrically rich goals for an inverse dynamics module, enabling precise and spatially-aware manipulation.

iii) Experiments demonstrate that our method outperforms strong video generative planning baselines in both generation quality and downstream robotic task performance, including strong zero-shot generalization to various visual changes, and effectiveness on real-world robot tasks.

## 2. Related Work

**General Purpose Embodied Models.** A dominant paradigm in robotic learning and embodied agents has been the development of large multitask policies that directly map sensory inputs to actions. Through the collection of large-scale multi-task embodied and robotics datasets, such models (Reed et al., 2022; Lee et al., 2022; Huang et al., 2023; Zitkovich et al., 2023; Kim et al., 2024; Barreiros et al., 2025; Hou et al., 2025; NVIDIA et al., 2025; Black et al., 2024) are able to solve tasks across many environments. However, there are two large challenges with constructing such general-purpose policies across many environments. First, the action space across environments is often misaligned, with existing works requiring careful action tokenization (Reed et al., 2022), and second, small changes in the environment cause policies to fail. To address these limitations, we instead learn a structured 4D latent predictive model and perform planning in latent space. By operating on a shared 3D state rather than directly predicting actions, our approach enables transfer across environments and robustness to environmental variations.

**Video Generative Models for Planning.** Learned video generative models have recently been explored for robot planning, often through video prediction from a single viewpoint (Janner et al., 2022; Ajay et al., 2023a; Li, 2023; Ajay et al., 2023b; Du et al., 2023a; Ko et al., 2023; Yang et al., 2023; Li et al., 2023; He et al., 2023; Alonso et al., 2024; Chen et al., 2024; Ubukata et al., 2024; Bar et al., 2025; Qi et al., 2025; Xie et al., 2025a). For example, UniPi (Du et al., 2023b) frames planning as generating a video trajectory, which improves interpretability but lacks explicit 3D structure, leading to inconsistencies under occlusion or viewpoint change. To address this, hybrid approaches such as TesserAct (Zhen et al., 2025) extend video models to predict future depth and normal maps, providing stronger spatial priors for manipulation. However, these methods are fundamentally 2.5D and operate in pixel space, which struggle to maintain full multi-view coherence. In contrast, our method models dynamics directly in a structured 3D

latent space, enabling consistent multi-view rollouts and geometrically grounded subgoals.

**3D Dynamics and Planning with Explicit Geometry.** A parallel line of research learns dynamics over structured 3D representations such as point clouds, meshes, NeRF-like fields, or 3DGS, enabling physical simulation or relational reasoning for manipulation tasks. These include behavior-primitive dynamics for stowing (Chen et al., 2023), point-cloud relational planning (Huang et al., 2025), deformable-object digital twins (Jiang et al., 2025; Zhang et al., 2025), and Gaussian-based 3D predictive models (Lu et al., 2024; 2025; Chai et al., 2025). Similarly, graph-based dynamics have been applied to elasto-plastic manipulation (Shi et al., 2023; 2022) and latent relational planners (Huang et al., 2024), while others utilize compositional NeRFs for multi-object scenes (Driess et al., 2023). While prior structured 3D dynamics methods are effective in specific domains, they often rely on object-centric factorizations, predefined primitives, or task-specific structures. In contrast, our approach learns a holistic latent 3D scene representation that supports 4D rollouts and planning in a unified framework, without requiring predefined object primitives or action parameterizations.

# 3. Formulation of Latent Predictive Modeling

## 3.1. Problem Formulation

Our goal is to build a 4D predictive model that learns the dynamics of a 3D environment over time. We formalize it as a conditional generator $g(\boldsymbol{o}_{t+1}, ..., \boldsymbol{o}_{t+T}|\boldsymbol{o}_t, c)$. Here, given the state of the 3D scene $\boldsymbol{o}_t$ at time $t$ and a high-level task instruction $c$, the model predicts a sequence of future 3D scene states $\{\boldsymbol{o}_{t+1}, ..., \boldsymbol{o}_{t+T}\}$ over a horizon $T$.

In practice, the complete 3D scene $\boldsymbol{o}_t$ is not directly observable. Instead, it is only seen through partial observations $\{o_t^{(i)}\}$, such as RGB or depth images from multiple cameras in real-world setups, or renderings from simulated viewpoints. These observations must be geometrically consistent, as they all describe the same underlying 3D structure $\boldsymbol{o}_t$. The action $a$ can range from a low-level control signal to a high-level semantic instruction. In this work, we focus on text-based instructions that specify the desired evolution of the agent and the environment. In our implementation, language instructions serve as the high-level action input that guides the latent rollout, while the inverse dynamics module produces the low-level robot commands as absolute joint positions.

Prevailing predictive models are primarily based on video generation, predicting sequences of 2D frames (sometimes augmented with depth and normals) from a single viewpoint $g^{(i)}(o_{t+1}^{(i)}, ..., o_{t+T}^{(i)}|o_t^{(i)}, c)$. A straightforward extension to 3D is to train separate predictive models for each viewpoint

and then fuse their outputs. However, such designs do not naturally support coherent 4D scene prediction. Instead, we introduce a *structured 4D latent predictive model* that directly addresses the key requirements:

- **3D consistency**: By encoding the complete 3D scene at timestep $t$ into a single holistic latent representation $z_t$, our model ensures that predictions across views adhere to the same underlying 3D structure.
- **Multi-view reasoning**: The shared latent aggregates information from multiple observations, allowing cues from one view to inform predictions in others.
- **Flexible generalization**: The latent can be decoded into diverse explicit 3D formats (e.g., point clouds, 3D Gaussians), allowing the framework to adapt to novel viewpoints and various scene representations.

Together, these properties enable a unified 4D latent predictive model that predicts future latent states, $g(z_{t+1}, ..., z_{t+T}|z_t, c)$. The latent $z_t$ is designed to be decodable into various explicit 3D representations, such as point clouds or 3D Gaussians, which allows us to obtain any desired observation $o_t^{(t)}$ decoded from the state.

## 3.2. Structured 3D Latent for Scene Representations

Our predictive model requires a 3D latent representation $z$ that is both compact enough for efficient dynamic modeling and expressive enough to capture the fine details of the complete 3D structure. Traditional representations, such as meshes, point clouds, or SDFs, often lack photorealism, while modern representations, like NeRFs (Mildenhall et al., 2021) or 3D Gaussians (Kerbl et al., 2023), are computationally expensive to generate directly at every timestep.

To balance efficiency and expressivity, we adopt a structured, sparse latent representation inspired by SLAT (Xiang et al., 2025). Our latent scene representation $z_t$, is defined as a set of sparse voxel features: $z_t = \{(\boldsymbol{p}_i, \boldsymbol{f}_i)\}_{i=1}^L$. Here, within a discretized $N \times N \times N$ grid of the 3D scene, $\boldsymbol{p}_i \in \{0, 1, ...N-1\}^3$ denotes the 3D coordinate of one of the $L$ active voxels, and $\boldsymbol{f}_i \in \mathbb{R}^d$ is a feature vector encoding local geometry and color. This representation balances structural information with latent compression. Compared to a standard dense 3D grid at resolution $N = 64$ requiring $64^3$ elements, our structured voxel latent uses a sparse set of approximately $L \approx 8000$ active voxels, each carrying a compact feature $d = 8$ in our settings.

**Encoding from images to structured 3D latent.** The encoding process uses calibrated multi-view RGB-D observations together with camera intrinsics and extrinsics. We first unproject the depth map using camera geometry and merge them into a 3D point cloud, which is then voxelized into a sparse grid $\{\boldsymbol{p}_i\}_{i=1}^L$. For each image, we use a pre-trained DINOv2 encoder to extract patch-level embeddings. The

*Figure 2.* **Structured 4D latent predictive model for robot planning.** The model reconstructs a 3D latent from multi-view images. The structured 4D latent predictive model then predicts future latents conditioned on the current state and a text instruction, using a Single Dynamics Model for coarse structural changes and a Latent Generator for detailed features. The predicted latents are decoded into explicit 3D formats such as point clouds or rendered views, which are subsequently used by a goal-conditioned inverse dynamics model to produce robot actions.

DINOv2 patch embeddings are unprojected into the voxel grid using the same camera poses, and the features arriving at the same voxel are averaged, then a latent encoder $\mathcal{E}$ to produce the latent features $\boldsymbol{f}_i$. This produces the structured 3D latents $z_t = \{(\boldsymbol{p}_i, \boldsymbol{f}_i)\}_{i=1}^{L}$.

**Decoding from structured 3D latent to images.** To get back to a renderable scene, a latent decoder $\mathcal{D}$ maps each latent voxel feature $\boldsymbol{f}_i$ to a set of $K$ 3D Gaussians $\{(\boldsymbol{o}_i^k, \boldsymbol{c}_i^k, \boldsymbol{s}_i^k, \alpha_i^k, \boldsymbol{r}_i^k)\}_{k=1}^{K}$. The final center of each Gaussian is constrained to remain near the active voxel by $\boldsymbol{x}_i^k = \boldsymbol{p}_i + \tanh(\boldsymbol{o}_i^k)$. The decoded 3D Gaussians can be rendered into images from arbitrary viewpoints, or be converted into a point cloud by taking the Gaussian centers as explicit 3D points. This establishes a mapping from the 3D latent $z_t$ to observation $o_t^{(i)}$.

We use the pre-trained 3D encoder $\mathcal{E}$ and decoder $\mathcal{D}$ from TRELLIS (Xiang et al., 2025), which was trained using RGB reconstruction losses (L1, D-SSIM, LPIPS) to supervise the 3D Gaussians. This encoder-decoder architecture bridges raw visual perception (2D images) and a structured internal 3D scene state ($z_t$). With this representation in place, the next step is to learn temporal dynamics in latent space.

## 4. Structured 4D Latent Predictive Model

We propose a *Structured 4D latent predictive model* to predict the dynamics of 3D scenes. The model generates future 3D structures conditioned on current observations and textual instructions. With the ability to model 3D dynamics, it can serve as a planner for robot manipulation tasks, and when combined with an inverse dynamics module, it converts predicted 3D futures into executable robot actions. An overview of the framework is shown in Figure 2.

### 4.1. Conditioned 3D Latent Sequence Generation

We formulate 4D latent prediction as a conditional generator in latent space: $g(z_{t+1}, ..., z_{t+T}|z_t, c)$, as detailed

in Section 3. The generator operates autoregressively $g(z_{t+1}|z_t, c)$, and future states are obtained by iterative rollout. Due to the complexity of generating a full 3D latent state at once, we adopt a two-step pipeline: a Single Dynamics Model $SD$ that forecasts the coarse geometry of the next state, and a Latent Generator $LG$ that fills in detailed feature representations. Together, they construct the next latent $z_{t+1}$, which is then fed forward for rollout.

**Data Preparation.** Each training sequence is represented as $(z_1, \ldots, z_T, c)$. For a robot task, we uniformly sample $T$ intermediate timesteps as subgoals. At each $t$, multi-view images are processed by a pre-trained encoder (Xiang et al., 2025) to obtain a 3D latent $z_t$. During training, we randomly choose $t \in \{1, \ldots, T-1\}$ and use $(z_t, z_{t+1}, c)$ pairs for supervision.

**Single Dynamics Model.** The single dynamics model $SD(\{\boldsymbol{p}_i\}_{t+1}|z_t, c)$ focuses on the dynamics, which predicts the sparse voxels of the next state conditioned on the current latent and the text instruction.

- **Modeling.** We use conditional flow matching (Lipman et al., 2023) for generative modeling , which is closely related and largely equivalent in formulation to standard diffusion/score-matching objectives. Here, we adopt flow matching for simplicity and consistency in our setup. The voxel grid $\{0, 1\}^{N^3}$ is first encoded by 3D convolutional blocks and compressed into a latent tensor $\mathbb{R}^{N_c^3}$ with lower resolution ($N_c < N$). A transformer denoiser then operates in this compressed space.

- **Conditioning.** Text instructions are encoded with a pre-trained CLIP model (Radford et al., 2021). The current latent $z_t$ is processed by 3D convolutions and aligned to resolution $N_c$. Both conditions share positional encodings with the voxel tokens, enabling the model to capture correlations, and are injected in each transformer block through cross-attention. To improve robustness to partial observations, we use condition augmentation, randomly dropping out voxel features from the input latent condition $z_t$ and adding Gaussian noise to its features $\{\boldsymbol{f}_i\}$.

**Latent Generator.** The latent generator $LG(\{\boldsymbol{f}_i\}_{t+1}|\{\boldsymbol{p}_i\}_{t+1}, z_t, c)$ predicts voxel features for the structure given by $SD$. Unlike $SD$, $LG$ focuses on appearance and visual details rather than dynamics. Similar to $SD$, it adopts a flow-matching framework with a transformer backbone, conditioned on text and 3D latent features via cross-attention. With this design, SD and LG can be trained separately but applied iteratively: SD predicts voxel positions, and LG fills in their features, producing complete 3D latents $z_{t+1}, ..., z_{t+T}$ over time.

**Training Objectives.** The Single Dynamics model (SD) and Latent Generator (LG) are trained independently as conditional flow-matching models in latent space. Concretely, for each component we optimize a conditional flow-matching objective of the form as follows:

$$\mathcal{L}_{\text{FM}}(\theta) = \mathbb{E}_{t,\epsilon,x_0} \left\| v_\theta(x(t), t, c) - (\epsilon - x_0) \right\|^2,$$

where $\theta$ (respectively, $\theta_{SD}$ and $\theta_{LG}$) denotes the model weights for $SD$ and $LG$, $x_0$ is the clean data sample, and $x(t) = (1 - t)x_0 + t\epsilon$ is the interpolation between $x_0$ and Gaussian noise $\epsilon \sim \mathcal{N}(0, I)$ at the sampling step $t \in [0, 1]$.

## 4.2. Planning with Inverse Dynamics

The structured 4D latent predictive model provides a high-level latent plan for robotic control. Given a text instruction $c$ and the current state latent $z_0$, the model predicts future states $z_1, \ldots, z_T$ describing how the agent will interact with the environment.

**Goal-Conditioned Inverse Dynamics.** To translate predicted latents into robot control, we employ a goal-conditioned inverse dynamics module $ID(\cdot)$ that maps a current state and a latent subgoal to a short-horizon command sequence: $ID(z_t, z_{t+1}) \rightarrow a_{1:H}$, where $a_{1:H}$ are desired joint positions that drive the robot from $z_t$ toward the subgoal $z_{t+1}$. We consider two implementations:

- **Learned inverse dynamics model.** We decode $z_t$ and $z_{t+1}$ into scene point clouds $pc_t$ and $pc_{t+1}$, encode them with a pyramid convolutional backbone (Ze et al., 2024a), concatenate the resulting features with the robot proprioceptive state, and use a diffusion head to predict $a_{1:H}$. The training objective is:

$$\mathcal{L}_{ID}(\theta_{ID}) = \mathbb{E}_{t,\epsilon,u_0} \| \epsilon_t - \epsilon_{\theta_{ID}}(u_0 + \epsilon_t, pc_t, pc_{t+1}, t)\|^2,$$

where $\theta_{ID}$ is the model weights, $\epsilon_t \sim \mathcal{N}(0, I)$ is random Gaussian noise, and $u_0$ is the clean action chunk $a_{1:H}$. This inverse dynamics model is trained separately from the predictive model.

- **Learning-free pose estimation with motion planning.** We also consider a learning-free inverse dynamics variant that converts predicted 3D subgoals into executable motions through geometric registration. We first decode

**Algorithm 1** Structured 4D Latent Predictive Model for Robot Planning

---
1: Observe initial multi-view images $\{o_0^{(i)}\}$.
2: Encode initial state: $z_0 \leftarrow \mathcal{E}(\{o_0^{(i)}\})$.
3: **for** $t = 0, ..., T - 1$ **do**
4:     Predict next latent subgoal based on current state $z_t$ and instruction $c$: $\hat{z}_{t+1} \leftarrow LG(SD(z_t, c), z_t, c)$.
5:     Decode latent to point cloud $pc_t \leftarrow \mathcal{D}(z_t)$ and $\hat{pc}_{t+1} \leftarrow \mathcal{D}(\hat{z}_{t+1})$.
6:     Predict action chunk with inverse dynamics module $a_{1:H} \leftarrow ID(pc_t, \hat{pc}_{t+1})$.
7:     Execute action chunk $a_{1:H}$.
8:     **if** closed loop planning **then**
9:         Observe new multi-view images $\{o_{t+1}^{(i)}\}$.
10:         Update latent state: $z_{t+1} \leftarrow \mathcal{E}(\{o_{t+1}^{(i)}\})$.
11:     **else**
12:         Set $z_{t+1} \leftarrow \hat{z}_{t+1}$.
13:     **end if**
14: **end for**

---

the predicted latent subgoal $z_{t+1}$ into a point cloud $pc_{t+1}$ expressed in the robot base frame. Given a template point cloud of the robot gripper $pc_{\text{gripper}}$, we estimate the rigid transformation that aligns the gripper template to the predicted gripper geometry in $pc_{t+1}$. Specifically, we compute FPFH features for coarse correspondence matching, use RANSAC to obtain an initial alignment, and refine the resulting transformation with ICP. The final transformation defines the target end-effector pose $T_{\text{ee},t+1} \in SE(3)$ in the robot base coordinate frame. We then pass $T_{\text{ee},t+1}$ to a motion planner, which generates a feasible trajectory for the robot to reach the predicted subgoal.

**Planning Pipeline.** Starting from initial multi-view observations, the predictive model predicts a sequence of latent subgoals $z_1, \ldots, z_T$ conditioned on the instruction $a$. At each step, a goal-conditioned inverse dynamics module translates the current latent state and the next subgoal $(z_t, z_{t+1})$ into a short-horizon action chunk, which is executed on the robot. The agent repeatedly replans until the subgoal is reached, then proceeds to the next subgoal. Optionally, in closed-loop execution, the latent state is updated from new observations after action execution. We summarize the planning pipeline in Algorithm 1.

## 5. Experiments

In this section, we evaluate our structured 4D latent predictive model. We first describe the experimental setup, then report 4D generation results that assess visual quality and 3D consistency, followed by downstream robot planning

*Table 1.* **Evaluation of 4D generation quality.** We collect 5 key frames per trajectory and 40 camera views per frame. We report (a) image and 3D consistency metrics and (b) robot mask IoU. Our method improves 3D consistency and mask accuracy over Wan-2.1, TesserAct, and OpenSora-1.3, benefiting from an explicit 3D latent representation.

| | PSNR↑ | SSIM↑ | LPIPS↓ | CD↓ | depth↓ | cPSNR↑ | cSSIM↑ | cLPIPS↓ | | StackCube | ToolPull | PegInsert | Avg. |
|---|---|---|---|---|---|---|---|---|---|---|---|---|---|
| Wan-2.1 | 19.87 | 0.84 | 0.09 | 43.09 | 25.06 | 16.86 | 0.62 | 0.24 | Wan-2.1 | 0.74 | 0.72 | 0.72 | 0.73 |
| TesserAct | 21.63 | **0.86** | **0.07** | 42.79 | 23.87 | 17.91 | 0.65 | 0.23 | TesserAct | 0.83 | 0.77 | 0.77 | 0.79 |
| OpenSora | 19.89 | 0.82 | 0.09 | 44.07 | 25.82 | 16.67 | 0.60 | 0.25 | OpenSora | 0.78 | 0.70 | 0.70 | 0.72 |
| Ours | **22.45** | 0.79 | 0.13 | **5.95** | **9.38** | **27.42** | **0.86** | **0.07** | Ours | **0.91** | **0.93** | **0.90** | **0.91** |

*(a)* **Image and 3D consistency.** We evaluate both standard image quality metrics (PSNR, SSIM, LPIPS) and 3D consistency metrics from MVGBench (Xie et al., 2025b) (Chamfer Distance, depth error, cPSNR, cSSIM, cLPIPS).

*(b)* **Robot mask IoU.** We evaluate the IoU between ground truth robot mask and generated robot mask.

results that test whether the predicted 3D structure enables effective manipulation. We conclude with ablations and real-world robot experiments that evaluate whether the approach can scale beyond simulation.

## 5.1. Experimental Setup

**Training Data for Structured 4D Latent Predictive Model.** We collect demonstration data from ManiSkill3 (Tao et al., 2025) and LIBERO (Liu et al., 2023) tasks, each paired with a language instruction. We generate 1,000 demonstrations per ManiSkill3 task and 50 per LIBERO-90 task. From each trajectory, we sample 4-10 intermediate timesteps and render 40 spherical multi-view cameras.

**Inverse Dynamics for Robot Planning.** To translate model predicted subgoals into executable actions, we train an inverse dynamics module (Sec. 4.2) on three ManiSkill3 tasks: StackCube-v1, PullCubeTool-v1, and PegInsertionSide-v1 (Tao et al., 2025). For each task, we collect 1,000 demonstration trajectories with point cloud observations aggregated from four cameras and the corresponding action sequences. We then evaluate robot planning on the same tasks under different random initial conditions. At test time, we consider both open-loop execution (execute the predicted action chunk) and closed-loop execution (replan after each action step using new observations). Unless otherwise specified, we use open-loop execution for efficiency, and report both settings in the ablation study.

**Baselines.** We compare our 4D generation and robot planning ability with the following baselines:

- **UniPi** (Du et al., 2023b), a video-based planner that uses inverse dynamics for control; we implement it by fine-tuning **Wan 2.1** (Wan et al., 2025).
- **TesserAct** (Zhen et al., 2025), a 4D generative model that generates future RGB, depth, and normals for manipulation.
- **OpenSora** (Zheng et al., 2024) is an image-to-video generation model, which is regarded as a baseline in video generation comparisons.

- **Diffusion Policy** (Chi et al., 2023) and **3D Diffusion Policy** (Ze et al., 2024b) are state-of-the-art imitation learning methods.

UniPi and TesserAct are fine-tuned on our dataset to generate task rollouts, which are converted to actions using an image-based diffusion inverse dynamics module. DP and DP3 are trained from expert demonstrations for each task.

## 5.2. 4D Generation Results

**Visual Quality.** Given initial multi-view images, our model autoregressively generates a sequence of future 3D latents to simulate the task's completion. Figure 3 visualizes the generated rollouts for several tasks. For each trajectory, we render the predicted 3D latents as images from two camera views and as a global point cloud. The results show that our generated 3D sequences maintain multiview consistency and temporal consistency while exhibiting high visual fidelity.

**Multiview Consistency.** Video generation-based models (UniPi, TesserAct, OpenSora) struggle to effectively integrate multi-view information. A common strategy for these models is to generate independent videos for each viewpoint and then attempt to fuse them at each timestep. However, without explicit 3D constraints, the independently generated views tend to lose consistency over time, which hinders the ability to leverage this multi-view information for downstream tasks, such as robot planning. In contrast, our model directly generates a unified 3D latent representation, which inherently enforces a consistent 3D structure and thus guarantees multi-view consistency by design. Table 1a shows that our method significantly outperforms fine-tuned Wan 2.1, TesserAct, and OpenSora 1.3 for multiview consistency.

**Viewpoint Generalization.** Many real-world tasks, particularly in mobile manipulation, cannot rely on fixed sensors and require observations from varying viewpoints. In such scenarios, it is crucial for a predictive model to generalize to novel viewpoints when simulating planning trajectories. Figure 4 demonstrates our model's robust ability to integrate diverse multi-view information and generalize to previously unseen viewpoints.

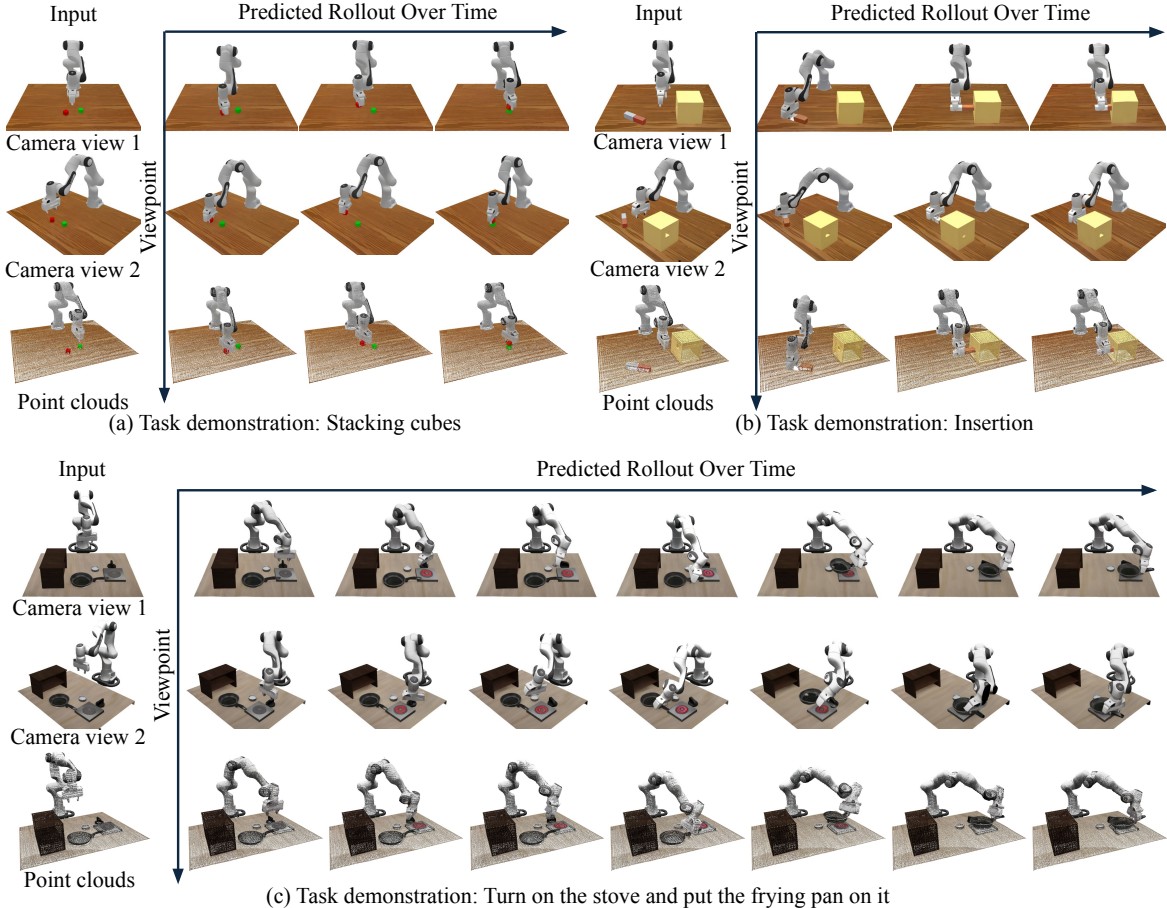

*Figure 3.* **4D generation visualizations.** Given input observations in the first column, our model unrolls the 4D latent dynamics to generate future 3D structures over time. For each subfigure, the first two rows show renderings from different camera viewpoints, and the third row shows corresponding point cloud visualizations.

*Table 2.* **Success rate for ManiSkill3 tasks.** Average success rate over 100 episodes, using four global cameras for observation. For PegInsertionSide-v1, the success clearance is relaxed to 0.01 m.

|  | StackCube | ToolPull | PegInsert* | Average |
|---|---|---|---|---|
| DP | 56% | 87% | **24%** | 55.7% |
| DP3 | 47% | **94%** | 7% | 49.3% |
| UniPi | 35% | 49% | 4% | 29.3% |
| TesserAct | 31% | 43% | 3% | 24.7% |
| Ours | **84%** | 84% | 16% | **61.3%** |

*Table 3.* **Success rate for RLBench tasks.** Average success rate over 100 episodes. For UniPi and TesserAct baselines, the success rate is reported in Zhen et al. (2025).

|  | Close Box | Sweep To Dustpan | Water Plants | Average |
|---|---|---|---|---|
| UniPi | 81% | 49% | 35% | 55.0% |
| TesserAct | 88% | 56% | 41% | 61.7% |
| Ours | **93%** | **69%** | **64%** | **75.3%** |

**Connection to Robot Policy Performance.** In order to further evaluate the connection of predictive rollout quality and robot policy performance more directly, we use Segment Anything Model 3 (SAM3) (Carion et al., 2025) to segment the robot shape mask in each generated image, and compare the IoU score of robot mask between generation and ground truth. Table 1b shows the IoU score of robot mask compared to the baselines, which shows that our proposed model can provide a more stable and accurate generation for robot action execution.

## 5.3. Robot Planning Results

**Manipulation planning performance.** We evaluate our structured 4D latent predictive model as a task planner, extracting actions at each step using a learned, goal-conditioned inverse dynamics model (Sec. 4.2). We compare our method's manipulation performance against video-generative planning baselines UniPi and TesserAct, and imitation learning policies DP and DP3. Table 2 shows that our method significantly outperforms the video-based predictive models and achieves performance comparable to

*Table 4.* **Zero-shot generalization with visual and viewpoint changes.** Success rates on the StackCube-v1 task under unseen test-time conditions. Perturbations include reduced lighting, additive Gaussian noise, background color shifts, and horizontal camera rotations ($5°$, $10°$).

| | Light | Noise | BG color | View ($5°$) | View ($10°$) |
|---|---|---|---|---|---|
| DP | 7% | 5% | 1% | 43% | 25% |
| DP3 | 47% | 47% | 47% | 49% | 45% |
| Ours | **78%** | **80%** | **84%** | **85%** | **83%** |

*Table 5.* **Ablation on inputs to the inverse dynamics module** on StackCube-v1 (success rate). The camera count indicates the views used to construct the 3D input for model training; inference always uses 4 cameras. OL/CL denote open-/closed-loop execution.

| Pointcloud-40 | | Pointcloud-4 | | 3D latent-4 | | Voxel-4 | |
|---|---|---|---|---|---|---|---|
| OL | CL | OL | CL | OL | CL | OL | CL |
| **84%** | **85%** | 57% | 84% | 66% | 80% | 57% | 73% |

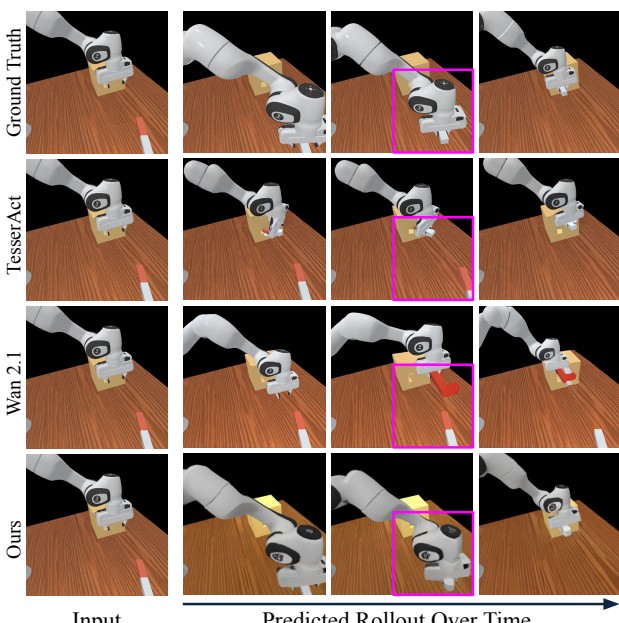

Figure 4 row labels: Ground Truth, TesserAct, Wan 2.1, Ours

Input — Predicted Rollout Over Time →

*Figure 4.* **Novel view generalization.** Models are trained on fixed global viewpoints and tested on a novel local viewpoint. As highlighted, baselines exhibit geometric inconsistencies and incorrect object interactions. Our method preserves consistent 3D structure and object placement from the unseen viewpoint.

the specialized imitation learning policies. Note that the original DP3 implementation does not use color information for better generalization ability, which prevents it from distinguishing between colored objects in the StackCube-v1 task.

We further evaluate on RLBench (James et al., 2020), where UniPi and TesserAct report strong results. We evaluated our method on three RLBench tasks: *CloseBox*, *SweepTo-Dustpan*, and *WaterPlants*, collecting 1,000 demonstrations for each. We utilized 20 cameras for model training and 4 cameras for inference, maintaining the same 4D predictive model and inverse dynamics architecture above. For the baselines, we cite the success rates reported in the official TesserAct publication (Zhen et al., 2025). Table 3 shows the success rate comparison in 3 RLBench tasks.

**Zero-shot generalization to visual and viewpoint changes.** Zero-shot generalization to novel visual conditions and cam-

era views is critical for deploying robotic policies in real-world, unstructured environments. Some recent works (Zhu et al., 2024) have mentioned this point with some studies explicitly evaluating robustness to such changes. As demonstrated in Section 5.2, our model uses an explicit 3D latent representation, which naturally provides robustness to viewpoint changes. We now evaluate the policy's zero-shot planning performance under various perturbations, including changes in lighting, background color, additive image noise, and camera viewpoint. The results in Table 4 show that our method maintains a high success rate across these visual changes, demonstrating strong zero-shot generalization ability.

### 5.4. Ablation Study

**Inputs to the Inverse Dynamics Module.** We ablate the input representation for inverse dynamics module, comparing decoded point clouds (ours), 3D latents, and 3D voxels. Due to the high computational cost for 3D latents encoding with large number of camera views, here we use 4 cameras for inverse dynamics module training. Table 5 reports planning success rates under both open-loop (OL) and closed-loop (CL) execution. Decoded point clouds consistently outperform voxel-based inputs and achieve performance comparable to latent-based representations, while being significantly more lightweight. Notably, even with only 4-camera inputs, closed-loop execution substantially improves success rates and already exceeds strong baselines.

We use decoded point clouds as the inverse dynamics input for the following reasons: (i) They preserve sufficient geometry for control while being significantly lighter than 3D latents. (ii) They decouple inverse dynamics from the predictive model, enabling modular training and better transfer across tasks. (iii) They support a simple learning-free pipeline (Sec. 4.2) to map subgoals to actions. This learning-free approach reduces reliance on additional action-labeled data and can make the planning pipeline more generalizable across tasks and robot setups where registration-based pose estimation is feasible.

**Number of camera views.** We train predictive models with 4, 10, and 40 camera views while keeping inference to 4 views. Planning success and 3D consistency improve with additional training views (Table 6a and Table 6b), but even the 4-view model substantially outperforms video-based

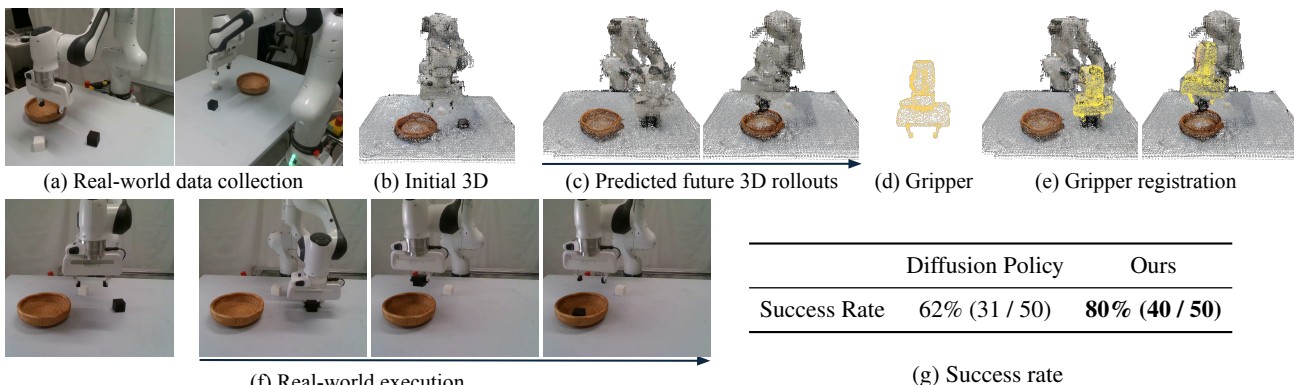

(a) Real-world data collection    (b) Initial 3D    (c) Predicted future 3D rollouts    (d) Gripper    (e) Gripper registration

(f) Real-world execution

|  | Diffusion Policy | Ours |
|---|---|---|
| Success Rate | 62% (31 / 50) | **80% (40 / 50)** |

(g) Success rate

*Figure 5.* **Real-world experiments.** From real robot observations (a), we reconstruct an initial 3D scene (b), and predict future rollouts (c). Given the gripper geometry (d), we register the predicted gripper trajectory to the reconstructed scene for execution (e) and run the policy on a real robot (f). Quantitative success rates are shown in (g).

baselines. Moreover, closed-loop replanning yields a large gain when training views are limited, making the 4-view setting particularly strong. This is consistent with replanning correcting imperfect rollouts under limited training-time multi-view supervision. Overall, the method remains effective with limited multi-view supervision, with closed-loop replanning providing the largest gains in this regime.

### 5.5. Real World Experiments

To evaluate the real-world applicability of our model, we collected a dataset of 200 human demonstrations for a block-in-basket task using four RGB-D cameras. We sampled four intermediate frames from each trajectory and processed them into a training dataset following the data preparation procedure in Section 4.1. Figure 5 (a) illustrates our data collection setup.

We trained the structured 4D latent predictive model on the collected real-world dataset with the same configuration as simulations. We adopt learning-free point cloud registration method to extract the end effector pose, and execute the motion planning to reach each intermediate subgoal. Figure 5 (b) and (c) illustrate qualitative generation results for real-world scenarios, and Figure 5 (f) presents visualizations of policy rollouts. The generated point cloud sequences exhibit temporal and 3D consistency, and the successful demonstrations indicate that our model learns meaningful dynamics from real-world data. We randomly initialize object positions and evaluate over 50 episodes. Figure 5 (g) shows that our method achieves a higher success rate than the baseline, demonstrating that our proposed approach is effective for real-world robotic manipulation.

## 6. Conclusion

We have introduced a *Structured 4D latent predictive model* for robot planning, which predicts the evolution of 3D scene

*Table 6.* **Ablations on training-time camera views** on StackCube-v1. Ours-$K$ denotes a predictive model trained with $K$ camera views; all methods use 4 cameras at inference time.

| Method | CD ↓ | depth ↓ | cPSNR ↑ | cSSIM ↑ | cLPIPS ↓ |
|---|---|---|---|---|---|
| Wan-2.1 | 38.74 | 24.54 | 16.95 | 0.62 | 0.22 |
| TesserAct | 39.11 | 24.86 | 17.75 | 0.64 | 0.22 |
| Ours-4 | 7.10 | **8.81** | **28.89** | **0.86** | 0.07 |
| Ours-10 | 7.06 | 9.85 | 26.92 | 0.85 | 0.07 |
| Ours-40 | **6.81** | 9.98 | 26.75 | 0.85 | **0.07** |

*(a)* **Visual consistency metrics.**

| Ours-40 | | Ours-10 | | Ours-4 | | DP | DP3 | UniPi | TesserAct |
|---|---|---|---|---|---|---|---|---|---|
| OL | CL | OL | CL | OL | CL | | | | |
| **84%** | **85%** | 72% | 82% | 57% | 84% | 56% | 47% | 35% | 31% |

*(b)* **Planning success rate.** OL/CL denote open-/closed-loop execution.

structure directly in a structured latent space. By moving beyond prevailing 2D video-based approaches, our model learns a dynamic model in 3D latent space that encodes holistic scene structure to enforce 3D consistency, producing rollouts of future latents that can be decoded into explicit formats such as point clouds or rendered views. Integrated with a goal-conditioned inverse dynamics module, these latents serve as geometrically grounded subgoals that translate into executable actions. Our experiments demonstrate that this approach achieves strong performance in 3D-aware generative modeling, yielding significant improvements in downstream robotic planning tasks. While our current implementation assumes calibrated multi-view inputs to reconstruct the initial latent, extending to weaker input settings is a promising direction for broader applicability.

## Acknowledgements

We acknowledge support from Kempner Institute, PickleRobotics, and Amazon AGI Labs.

## Impact Statement

This paper presents work whose goal is to advance the field of Machine Learning. There are many potential societal consequences of our work, none which we feel must be specifically highlighted here.

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

# A. Project Website and Additional Visualizations

To facilitate inspection of qualitative results, we provide an interactive project website at https://structured-4d-model.github.io/. The website contains higher-resolution visualizations, additional rollouts, and interactive renderings that are difficult to include in the paper due to space constraints.

**Robot planning visualizations (text-conditioned rollouts).** We evaluate text-conditioned generation results by applying multiple LIBERO-90 tasks to the *same* input scene. For each instruction, we unroll our structured 4D latent predictive model to generate future 3D Gaussian-splatting states and render them from a camera that sweeps along a circular trajectory. These visualizations highlight that the predicted 4D rollouts remain geometrically consistent while adapting to different task instructions.

**Real-world experiments.** We include three real-world execution rollouts of the full planning pipeline showing the robot execution for the block-in-basket task.

# B. Implementation Details

### B.1. Structured 4D Latent Predictive Model Training

Our framework consists of two main components: a single dynamics model and a latent generator, which were trained independently. Both were implemented as conditioned flow matching models following the architecture proposed by Xiang et al. (2025). Here, we extend the original conditions to both text and 3D latent. The latent generator operates on a $64 \times 64 \times 64$ voxelized grid with a feature dimension of $d = 8$, while the dynamics model uses a similar architecture on a coarser $16 \times 16 \times 16$ grid. Both models consist of 24 transformer blocks, each with 16 attention heads and a model dimension of 1024. For text conditioning, we utilized embeddings from a pre-trained CLIP model. The dynamics model is also conditioned on the input 3D latent; we use sparse 3D convolutional layers to match the latent's resolution to the model's internal dimension, after which it is injected into the cross-attention blocks together with the text condition.

Training for both models spanned 300,000 steps with a learning rate of $1 \times 10^{-4}$. We used a per-GPU batch size of 8 with mixed-precision (FP16) computation. For the latent generator, we applied 4 gradient accumulation steps, and for the dynamics model, we used 2 steps. The optimizer used was AdamW with no weight decay. To enable classifier-free guidance, we set the unconditional dropout probability to 0.1 and applied an exponential moving average (EMA) with a decay rate of 0.9999 to stabilize the training. Each model was trained for approximately 3 days on four NVIDIA H100 (80GB) GPUs.

### B.2. Inverse Dynamics Model Training

Our goal-conditioned inverse dynamics model is trained on 1,000 expert demonstrations for each task. The policy is formulated as a diffusion model that takes point clouds representing the current and goal states as input. For observation encoding, we employ a point cloud encoder introduced by Ze et al. (2024a), which comprises four 1D convolutional layers with a hidden dimension of 128. The action decoder is a 1D conditional UNet with downsampling channel dimensions of [256, 512, 1024], a kernel size of 5, and 8 groups for normalization layers.

The inverse dynamics model was trained for 20,000 epochs. At inference time, we use 100 denoising steps to predict the action sequence. Training for each task took approximately 8 hours on a single NVIDIA A100 GPU.

### B.3. Video Generative Baseline Fine-tuning

To establish our baselines, we utilize Wan 2.1 (Wan et al., 2025) as the video generative backbone of UniPi (Du et al., 2023b), and the TesserAct (Zhen et al., 2025) generative model. Both models were fine-tuned on the same dataset. For the TesserAct model specifically, we generated depth maps alongside the images from the simulator and used its official codebase to create normal maps. For fine-tuning, Wan 2.1 was trained for 10,000 steps on two NVIDIA H100 GPUs, achieving convergence in approximately 36 hours. TesserAct was also fine-tuned for 10,000 steps on four NVIDIA H100 GPUs, taking around two days to converge.

### B.4. Robot Planning Simulations

We evaluate each ManiSkill3 task over 100 episodes with fixed random seeds, using four $512 \times 512$ RGB cameras. For our method, DP, and DP3, we use the four views jointly as multi-view inputs. For video-generation-based baselines (UniPi and

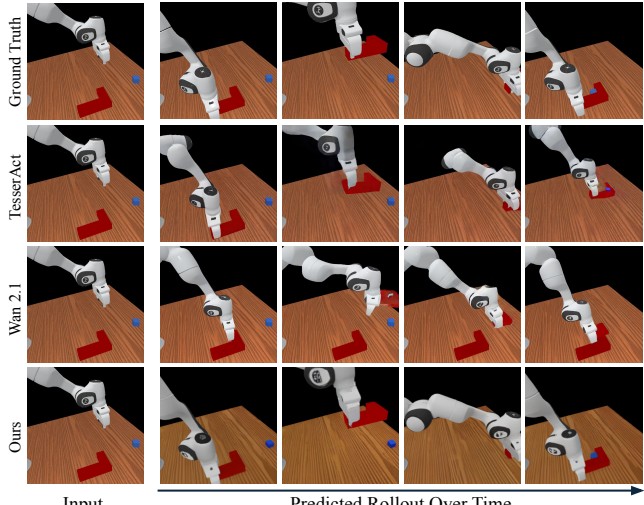

*Figure 6.* **Additional visualization on novel view generalization.** All models were trained on fixed global views but tested on a novel local viewpoint. Our model generates a consistent 3D scene from an unseen view, outperforming baselines significantly.

TesserAct), naively generating one video per view and then fusing them leads to severe multi-view inconsistency that can destabilize planning. Instead, we run inference four times, each conditioned on a single camera view, and count an episode as successful if any of the four single-view trials succeeds. Although TesserAct predicts depth and normal maps in addition to RGB, we found that the fused point clouds are often too noisy for reliable point cloud-based inverse dynamics; therefore, for this baseline we adopt the same image-based diffusion inverse dynamics module as UniPi.

For RLBench, we report UniPi and TesserAct numbers directly from TesserAct (Zhen et al., 2025). For our method, we follow the same evaluation protocol as in ManiSkill3 (four cameras at $512 \times 512$ and 100 fixed-seed episodes).

## C. Additional Experimental Results

### C.1. Additional Viewpoint Generalization Results

Figure 4 demonstrates that our method generalizes better than video generation based baselines under camera viewpoint shifts on ManiSkill3 PegInsertionSide-v1. We include an additional viewpoint generalization example on ManiSkill3 PullCubeTool-v1 in Figure 6.

### C.2. Additional Robot Planning Results

To strengthen the empirical evaluation of 3D-aware methods, we additionally compared with 3D Diffuser Actor (Ke et al., 2024) as a strong 3D policy baseline on ManiSkill3 StackCube-v1 task using the same collected trajectories in Table 7[1]. This baseline is competitive in the original setting, but our method is substantially more robust under stronger perturbations. We believe this supports the claim that predictive 3D rollout provides a meaningful robustness advantage beyond standard 3D policy learning.

*Table 7.* **Additional robot planning results with zero-shot generalization.** Success rates on the StackCube-v1 task under unseen test-time conditions. Perturbations include reduced lighting, additive Gaussian noise (noise level 0.05, 0.08, 0.10), background color shifts, and horizontal camera rotations ($10°$).

|  | Original | Light | Noise (0.05) | Noise (0.08) | Noise (0.10) | BG Color | View ($10°$) |
|---|---|---|---|---|---|---|---|
| Ours-CL | **91%** | 90% | **86%** | **90%** | **86%** | **89%** | **88%** |
| 3D Diffuser Actor | 90% | **91%** | 85% | 48% | 7% | 78% | **88%** |

---

[1]For this comparison, we set the maximum episode length to 300 steps for both Ours-CL and 3D Diffuser Actor; the main-table ManiSkill3 results use a 200-step limit.

# D. Failure Modes and Limitations

### D.1. Failure Modes

Failures mainly arise from two sources: (1) **3D rollout errors** in the predicted future 3D latents, which are most visible in fine-grained, high-precision interactions (e.g., peg insertion), where small geometric deviations can lead to contact mismatch; and (2) **control errors** from the goal-conditioned inverse dynamics module, including geometry registration error for learning-free module and action prediction drift for learned inverse dynamics model.

### D.2. Limitations

While our framework demonstrates the benefit of structured 3D predictive rollouts for robot planning, it has several limitations. First, the current formulation relies on calibrated multi-view observations to reconstruct the initial 3D latent state, which may limit applicability in settings with sparse, monocular, or poorly calibrated sensing. Second, although the real-world results are encouraging, our validation is currently limited to a tabletop manipulation setup, and broader evaluation across more diverse robots, environments, and task categories remains future work. Finally, the method is sensitive to fine geometric precision, especially for contact-rich manipulation tasks such as peg insertion, where small reconstruction or prediction errors can lead to execution failure.

