# OpenReview forum: "Structured 4D Latent Predictive Model for Robot Planning"
_ICML.cc/2026/Conference — ICML 2026 regular_

### Official Review · Reviewer_m3jF · 2026-03-12

**Soundness:** 3
**Presentation:** 3
**Significance:** 4
**Originality:** 2
**Overall Recommendation:** 5
**Confidence:** 3

**Summary:**

This paper proposes a world model which includes a structured 3D voxel-based representation from SLAT [1], a single-step dynamics model based on these 3D latent representations, and a latent generator which reconstructs the 3D structured features to either point cloud format, or Gaussians that render RGB images. An inverse dynamics model is trained on the latent 3D features to extract actions. Results show consistent improvements in manipulation benchmarks against non-3D based world models’ performances.

[1] “Structured 3D Latents for Scalable and Versatile 3D Generation”, CVPR 2025.

**Compliance With Llm Reviewing Policy:**

Affirmed.

**Final Justification:**

The paper is technically sound and presents convincing empirical results, especially in terms of robustness and generalization to new viewpoints, which are very important aspects for real-world deployment. I still find originality to be somewhat limited, but the overall contribution to be sufficient. I disregarded my main concern regarding comparisons with Gaussian-based world models, as these are concurrent works, and fair comparisons are non-trivial within the rebuttal timeline. The authors clarified several aspects of their work and indicated they will improve clarity in the final manuscript. I increased my score accordingly.

**Key Questions For Authors:**

- Can you explain the encoding process more in detail? From Figure 2, it seems that the encoding process requires multi-view RGBD images and their respective camera parameters, since only surface level points are shown. Is this just a neat visualization or did you simply project patch embeddings along every voxel on the ray (i.e., no use of depth images)?
- For the decoding process, each latent voxel feature produces a set of K 3D Gaussians. I’m assuming point cloud is produced by using only the mean vectors of these Gaussians. How exactly did you produce multiple points from a single latent voxel feature?
- Can you provide results on additional RLBench tasks, for a direct comparison to Gaussian world models (e.g., GAF provides RLBench results in Table 1. [1])? If not, could you provide explanations as to why you omitted such results?

[1] “GAF: Gaussian Action Field as a 4D Representation for Dynamic World Modeling in Robotic Manipulation”, ICRA 2026.

**Limitations:**

Requires multiple viewpoints that cover every surface of the given scene, and their respective camera poses. Thus, such a world model is applicable only in controllable and isolated environments.

**Strengths And Weaknesses:**

**Strengths:**

- The authors discuss and explore the use of 3D representations for better physical consistency in world models for manipulation tasks, and empirical results show their superiority over video based world models.
- The paper is well structured and high level ideas are easy to follow.

**Weaknesses:**

- Limited novelty. Most components are already high performing models that are simply reused (and finetuned). The combination of a latent world model and an inverse dynamics model to extract a policy has been thoroughly explored before.
- While the authors mention closely related work, specially 3D based Gaussian world models [1-3], they do not show direct comparisons against those.

[1] “GWM: Towards Scalable Gaussian World Models for Robotic Manipulation”, ICCV 2025.\
[2] “ManiGaussian: Dynamic Gaussian Splatting for Multi-Task Robotic Manipulation”, ECCV 2024.\
[3] “GAF: Gaussian Action Field as a 4D Representation for Dynamic World Modeling in Robotic Manipulation”, ICRA 2026.

---

> ### Author Rebuttal · Authors · 2026-03-31
>
> We thank the reviewer for the thoughtful feedback and for recognizing the technical solidity and potential impact of the paper. We address the main points below and will make these distinctions and implementation details more explicit in the final version.
>
> 1. **Novelty and contribution**. We claim that our main contribution is proposing a predictive planning framework in **structured 3D latents** rather than 2D pixel latents. Specifically, we model the world dynamics in a compact 3D latent space that is efficient for conditioning and generation, and that can be rolled out to produce future 3D scene predictions conditioned on text instructions. In contrast to prior latent world models mostly based on 2D video generation methods, our proposed method inherently models the dynamics directly in 3D latent spaces, where the predictions are more geometrically accurate and more 3D consistent for downstream action translation. Additionally, our framework disentangles future 3D prediction from robot action execution. In particular, the decoded explicit 3D point clouds enable a training-free motion-planning based inverse dynamics method to aquire executable actions from the generated outputs. We think this modular design of prediction and execution can improve robustness and generalization.
>
> 2. **Encoding details**. The encoding process uses calibrated multi-view RGB-D observations together with camera intrinsics and extrinsics. We first unproject the depth map using camera geometry and merge them into a 3D point cloud, which is then voxelized into a sparse grid. For each image, DINOv2 patch embeddings are unprojected into the voxel grid using the same camera poses, and the features arriving at the same voxel are averaged before being passed through the latent encoder. This produces the structured 3D latents $z_t=\{ (p_i, f_i) \}_{i=1}^L$. Thus, the multi-view fusion is geometry-aware and does use depth, and we are not simply projecting image features along rays without 3D supervision. We will revise the method description and update the figure to explicitly include camera poses for clarity.
>
> 3. **Decoding details**. In our implementation, the point cloud is obtained from the decoded Gaussian centers. Concretely, following TRELLIS, each latent voxel $z_t=\{ (p_i, f_i) \}_{i=1}^L$ is decoded by the Gaussian decoder into $K$ local Gaussians, each with an offset, color, scale, opacity, and rotation. The final center of each Gaussian is constrained to remain near the active voxel by $x_i^k = p_i + \tanh (o_i^k)$. Therefore, a single latent voxel can produce $K$ 3D Gaussians, and the decoded point cloud is formed by taking these Gaussian centers as explicit 3D points.
>
> 4. **Gaussian-based 3D world models baselines and comparison**. We thank the reviewer for pointing out the related works. We will add the discussion of them in the related work section. We agree that a broader direct empirical comparison would be valuable. However, because GWM and GAF are concurrent works and have not released the full training and evaluation code for reproduction, a comparison under the same evaluation settings (e.g. camera viewpoints and training trajectories) is currently infeasible. ManiGaussian is a related method, but its training and evaluation protocol differ from ours in task set, camera placements, number of training demos, and evaluation views. Thus, a fair comparison would require substantial re-engineering and is infeasible within the rebuttal timeline. If the reviewer finds it valuable, we would be happy to include the comparison in the final version of the paper.
>     To strengthen the empirical evaluation of 3D-aware methods, we additionally compared with 3D Diffuser Actor as a strong 3D policy baseline on ManiSkill3 StackCube-v1 task using the same collected trajectories. This baseline is competitive in the original setting, but our method is substantially more robust under stronger perturbations. We believe this supports the claim that predictive 3D rollout provides a meaningful robustness advantage beyond standard 3D policy learning.
>     |  |Original|Light|Noise (0.05)|Noise (0.08)|Noise (0.10)|BG Color|View (10$^\circ$)|
>     |---|---|---|---|---|---|---|---|
>     |Ours (CL)|91%|90%|86%|90%|86%|89%|88%|
>     |3D Diffuser Actor|90%|91%|85%|48%|7%|78%|88%|
>
> Overall, we appreciate the reviewer’s comments. We will use them to sharpen the positioning of the contribution, and to make the encoding/decoding pipeline more explicit in the final version of the paper.

---

> > ### Author Rebuttal · Reviewer_m3jF · 2026-04-04
> >
> > All of my concerns have been addresssed.

---

> > > ### Author Response · Authors · 2026-04-05
> > >
> > > Thank you for your kind response and for confirming that our rebuttal has addressed your concerns. We appreciate your time, careful evaluation, and expertise throughout the review process.

---

### Official Review · Reviewer_8LEU · 2026-03-12

**Soundness:** 3
**Presentation:** 3
**Significance:** 3
**Originality:** 3
**Overall Recommendation:** 5
**Confidence:** 3

**Summary:**

This paper proposes a structured 4D latent world model for robot planning. It builds a uniform sparse 3D latent scene representation from multi-view observations and predicts future latent states conditioned on language. The latent representation can be decoded into explicit 3D outputs such as point clouds or rendered views. These future states are then used as subgoals for a learned inverse dynamics model.

In the experimental section, the paper shows clearly improved multi-view consistency over video-based world-model baselines, competitive downstream planning results on simulated manipulation tasks, some robustness to perturbations, and a small real-world block-in-basket demonstration.

**Compliance With Llm Reviewing Policy:**

Affirmed.

**Key Questions For Authors:**

Please see weaknesses

**Limitations:**

Yes.

**Strengths And Weaknesses:**

### Strength

- The paper addresses an important problem: world models for robot planning should ideally reason in a spatially consistent 3D space rather than per-view image space. This is well motivated, and the use of sparse voxel latents is a sensible design choice that balances structure and efficiency.

- The paper shows large gains in multi-view consistency metrics and robot-mask IoU over video-based baselines, which directly support the motivation for structured 3D latent modeling.

- The empirical evaluation is broad. Beyond generation quality, the paper includes downstream planning, robustness to reduced camera views, ablations on execution mode, and a real-world robot experiment.

- The paper is well organized. The pipeline is easy to follow, and the connection between generative prediction and downstream control is well explained.

### Weaknesses

- The framework is strongest when trained with rich multi-view supervision, which may limit applicability in settings with fewer cameras or less controlled sensing.

- Real-world validation is still limited. The real-robot experiment is promising, but it is restricted to a single block-in-basket task with 200 demonstrations and 50 evaluation episodes.

- The camera ablation is informative, but it also shows that performance with fewer training views improves substantially in closed-loop mode. That suggests long-horizon open-loop rollouts are still not consistently reliable when multi-view supervision is limited.

---

> ### Author Rebuttal · Authors · 2026-03-31
>
> We thank the reviewer for the positive assessment and for the thoughtful comments. We respond below to the scope-related points and will clarify them more precisely in the final version.
>
> 1. **Multi-view supervision and closed-loop replanning**. The reviewer is right that the framework is strongest when trained with more multi-view supervision. At the same time, the camera count ablation suggests that, even when the number of training-time views is reduced, the method still remains effective and continues to outperform the video-based baselines. The large gain from closed-loop execution in the low-view case is also informative, which suggests that when training-time multi-view supervision is limited, re-observing the scene and replanning from the updated 3D state is especially valuable for correcting imperfect long-horizon rollouts. We will clarify this point more explicitly in the final paper.
>
> 2. **Real-world experiments**. We also agree that the current real-world validation is still limited. We view this current experiment as the evidence for the feasibility of real-world transfer: the model learns meaningful 3D predictive dynamics from real-world demonstrations, and the predicted 3D subgoals can be executed through a learning-free registration and motion-planning interface, achieving higher success than the baseline. We think that extending this to more tasks, more diverse objects, and less controlled environments is an important direction for future work. We also consider the proposed framework has the potential to generalize more broadly, since in real-world settings, an embodied agent can actively explore the new viewpoints to obtain better 3D reconstructions.
>
> We thank the reviewer again for the supportive and careful reading. We appreciate that the review captures the main contribution well, and we will use these comments to present the scope and limitations more clearly.

---

> > ### Author Rebuttal · Reviewer_8LEU · 2026-04-03
> >
> > Thanks for the rebuttal.

---

> > > ### Author Response · Authors · 2026-04-05
> > >
> > > Thank you for your positive feedback and for your recognition of our work. We sincerely appreciate your time, effort, and expertise throughout the review process.

---

### Official Review · Reviewer_uLdK · 2026-03-12

**Soundness:** 2
**Presentation:** 3
**Significance:** 2
**Originality:** 2
**Overall Recommendation:** 3
**Confidence:** 4

**Summary:**

This paper is about robotic action planning. The authors propose a world model that allows to decode several different 3D formats and is trained by predicting future scene configurations.  Input images are first encoded by DinoV2. The resulting features are lifted to 3D to yield a voxel representation.

The proposed Single Dynamics Module encodes a voxel representation of the scene given the predicted voxel representation and text as input. Images are first compressed by a 3D convolutional net. The resulting tokens (or voxels according to line 194, right column) are perturbed by noise. The authors also apply a dropout noise. Another module, the Latent Generator, predicts features for each voxel.
The actions are predicted by processing the 3D points (or voxels) at times t and t+1 with another convolutional net. A diffusion net then predicts the action.The authors also mention another implementation based on RANSAC and ICP. In the experiment section, they do not mention which implementation they use.

**Compliance With Llm Reviewing Policy:**

Affirmed.

**Key Questions For Authors:**

Please define the losses you used.

**Limitations:**

yes

**Strengths And Weaknesses:**

- The losses used to train the system are nowhere defined.

- Using future states as training targets is not particularly novel. See for instance:
[1] Tian at el., Predictive Inverse Dynamics Models Are Scalable Learners For Robotic Manipulation, ICLR 2025
While this paper directly predicts pixels, the conceptual difference from pixels to voxels is not very large.

- Experimental evaluation, table 1: I find the comparison with diffusion models such as WAN or OpenSora not fair. These models have never been trained on the data used in this paper. In other words, they are evaluated in a zero-shot setting. A small fine-tuning that includes some 3D losses would have probably closed the gap for CD and depth.

- Lines 155-162, right column: For the 3D lifting, how is this done? Is a matching first computed across the original images for the 3D reconstruction?

- Table 1, “We collect 5 key frames per trajectory and 40 camera views per frame.”: How are the key frames selected? Is the voxel representation only based on the keyframes?

---

> ### Author Rebuttal · Authors · 2026-03-31
>
> We thank the reviewer for the careful reading and for highlighting several places where the technical setup should be stated more explicitly. We address the main concerns below and will revise the paper to make these points clearer in the final version.
> 1. **Training objectives**. We thank the reviewer for pointing out this. We clarify that the framework contains three trainable components with separate roles. The Single Dynamics model (SD) and Latent Generator (LG) are trained independently as conditional flow-matching models in latent space. Concretely, for each component we optimize a conditional flow-matching objective of the form $\\mathcal{L}\_{FM}(\\theta)=\\mathbb{E}\_{t,\\epsilon,x_0} \\Vert v\_\\theta(x,t,c)-(\\epsilon-x\_0)\\Vert^2$, where $\theta$ (repsectively, $\theta_{SD}$, $\theta_{LG}$) denotes the model weights for SD and LG, $c$ represents the conditions (text and current 3D latents), and $x(t) =(1-t) x_0+t\epsilon$ is the interpolation between clean data sample $x_0$ and Gaussian noise $\epsilon \sim \mathcal{N}(0,I)$ at the sampling step $t\in[0, 1]$. For SD and LG, $x_0$ is the VAE latents of the sparse voxels $\{p\_i\}$ and features $\{f_i\}$, respectively.
> The learned inverse dynamics module (ID) is trained separately as a diffusion policy over action chunks $\\mathcal{L}\_{ID}(\\theta_{ID})=\\mathbb{E}\_{t,\\epsilon,a\_0}\\Vert\\epsilon_t-\\epsilon\_{\\theta\_{ID}}(a\_0+\\epsilon_t,o,t)\\Vert^2$, where $\theta_{ID}$ is the model weights for ID, $o$ is current and future point clouds, $\epsilon_t\sim\mathcal{N}(0,I)$ is random Gaussian noise, and $a_0$ is the action trajectory from current to goal.
> The TRELLIS 3D encoder-decoder is pre-trained and reused, so its reconstruction losses are not optimized during our world-model training.
> 2. **Novelty and contribution**. We would like to clarify that our main contribution is proposing a predictive planning framework in **structured 3D latents** rather than 2D pixel latents. Specifically, we propose a framework that can rollout and generate the sequence of future 3D latents, which can be decoded into explicit 3D Gaussians or point clouds. Compared to 2D video generation methods, which can be ambiguous about depth and precise spatial alignment (for example, the gripper may look aligned with the object in the 2D image, while still having an offset in 3D space), these 3D representations provide more accurate geometry for translating predictions into executable robot actions. Additionally, our framework disentangles future 3D prediction from robot action execution. In particular, the decoded explicit 3D point clouds enable a training-free motion-planning based ID method to acquire executable actions from the generated outputs. We believe the modular design of prediction and execution can improve robustness and generalization.
> 3. **Baseline protocol and fairness of comparison**. We thank the reviewer for this concern. We would like to clarify that the comparison in Table 1 is not zero-shot for the video generation baselines. Wan-2.1, TesserAct, and OpenSora are all finetuned on the same task trajectories as our method. However, because these video-generation based methods inherently model distributions in pixel space, they still fail to maintain 3D consistency across viewpoints even after finetuning.
> As for finetuning these baselines with additional 3D losses, we found that there is no straightforward implementation path. Since these baselines are fundamentally based on 2D video generation models and do not natively take multiview inputs, directly incorporating the 3D losses will be nontrivial.
> 4. **3D lifting details**. We start from calibrated multi-view RGBD observations. We first unproject the depth map using camera intrinsics and extrinsics and merge them into a 3D point cloud, which is then voxelized into a sparse grid. For each image, DINOv2 patch embeddings are unprojected into the voxel grid using the same camera poses, and the features arriving at the same voxel are averaged, then are passed through the latent encoder. This produces the structured 3D latents $z_t=\{(p_i,f_i)\}_{i=1}^L$.
> 5. **Keyframe protocol**. For Table 1, we uniformly sample 5 keyframes from each trajectory, and render multiview images for each keyframe. The latent voxel representation is constructed only from those selected keyframes.
> 6. **Inverse dynamics implementation clarification**. Both implementations of the ID module are effective in our framework. In simulation experiments (ManiSkill3 and RLBench), we use the learned diffusion-based ID module for more accurate control. In real-world experiment, because the point clouds are noisier, we instead use the training-free pipeline based on point-cloud registration and motion planning (FPFH/RANSAC/ICP) which is more robust.
>
> Overall, we appreciate the reviewer's suggestion, and we will revise the manuscript accordingly so that these technical details are stated more explicitly and clearly in the final verion.

---

### Official Review · Reviewer_qiSB · 2026-03-13

**Soundness:** 3
**Presentation:** 2
**Significance:** 2
**Originality:** 2
**Overall Recommendation:** 4
**Confidence:** 3

**Summary:**

This paper proposes a structured 4D latent planning pipeline for robot manipulation. Multi-view images are encoded into a sparse structured 3D latent scene representation, a dynamics model predicts future latent scene states conditioned on the current scene and a text instruction, and a goal-conditioned inverse dynamics module converts the predicted future states into executable robot joint trajectories. Experiments on ManiSkill3, RLBench, and a real-world manipulation setup show stronger 3D consistency than video-based baselines and competitive downstream planning performance.

**Compliance With Llm Reviewing Policy:**

Affirmed.

**Final Justification:**

The main concern about positioning will be resolved if the authors make the updates accordingly.

**Key Questions For Authors:**

See weaknesses

**Limitations:**

yes

**Strengths And Weaknesses:**

Strengths

- The problem is meaningful. Existing video-based planning / world-model approaches are fundamentally limited by their 2D representation, so moving toward a more explicit 3D-aware latent scene representation is a sensible direction for manipulation planning.

- The paper combines a structured sparse 3D latent representation, a two-stage latent dynamics model (coarse structure + latent feature generation), and a goal-conditioned inverse dynamics module into a reasonably complete planning system. The technical pipeline is coherent.

- The experimental section is fairly broad. The paper includes 4D generation metrics, downstream robot planning on ManiSkill3 and RLBench, zero-shot visual perturbation tests, ablations on camera count and inverse-dynamics inputs, and a real-world experiment.

Weaknesses

- My main concern is about positioning. I am not fully convinced by the “world model” framing in the classical model-based control sense. The learned dynamics model predicts future scene states conditioned on the current latent and a high-level text instruction, while the actual robot control interface is produced later by a separate inverse dynamics module. In that sense, the overall pipeline looks closer to goal-conditioned policy over predicted future states than to an action-conditioned world model. Related works such as [1] explicitly positions a similar paradigm as a *policy* rather than as a world model. I think the paper would be stronger if it clarified this distinction more carefully.

- The current contribution is better understood as a 3D-consistent planning pipeline than a fundamentally new world-model formulation. The overall design is reasonable, but the comparisons are still centered on video-based planners and imitation policies. More directly related 3D structured dynamics / planning directions already exist [2-6]. The paper would be stronger if it more clearly differentiated itself from these directions, or discussed more concretely why direct empirical comparison is not feasible. Relatedly, the main empirical advantage is most clearly on 3D consistency metrics; the claim of “superior visual quality” in a broad sense is less clear, since some standard 2D image metrics are weaker than the video-based baselines.

- It is still unclear how essential the latent representation itself is for downstream control. In practice, planning does not use the learned latent directly, but decoded point clouds. The ablation suggests that point clouds are competitive with, or sometimes better than, latent-based inputs for inverse dynamics in the reported setting. This does not invalidate the method, but it makes the practical role of the learned latent representation in control somewhat indirect.

- The method assumes calibrated multi-view inputs, the strongest results rely on relatively dense training-time multi-view supervision, and the inverse dynamics module is trained task-specifically on benchmark-specific demonstrations. The real-world evaluation is encouraging, but still limited to a single task. So the paper does support better 3D-consistent planning under this setting, but the evidence for broader generality is still somewhat limited.

Overall, I think the technical direction in this paper is reasonable, and I am convinced that moving from 2D video prediction to a structured 3D scene representation is beneficial for manipulation planning. My main concern is that the current contribution is best understood as a strong 3D-consistent planning pipeline under a fairly instrumented multi-view setting, and that the world-model positioning is somewhat overstated. I would recommend positioning the paper more carefully around this narrower but still meaningful contribution.


[1] Learning Universal Policies via Text-Guided Video Generation.
[2] Points2Plans: From Point Clouds to Long-Horizon Plans with Composable Relational Dynamics.
[3] Latent Space Planning for Multi-Object Manipulation with Environment-Aware Relational Classifiers.
[4] GWM: Towards Scalable Gaussian World Models for Robotic Manipulation.
[5] 3D Diffuser Actor: Policy Diffusion with 3D Scene Representations.
[6] Learning Multi-Object Dynamics with Compositional Neural Radiance Fields.

---

> ### Author Rebuttal · Authors · 2026-03-31
>
> We thank the reviewer for the thoughtful and constructive feedback. We respond below to the main concerns and will clarify these points more carefully in the final version.
>
> 1. **Positioning and world model framing**. We appreciate the reviewer’s distinction between classical action-conditioned control and our setting. Specifically, our paper proposes a predictive model of future 3D world states for planning: given current multiview observations and a text instruction, the model rolls out future 3D latent scene states, which can be decoded to explicit 3D point clouds and translated into executable robot actions. In this sense, our model is also learning the dynamics of the world and predicting future states based on the current state and conditions, which aligns with the broad notion of world models. That said, we fully agree that we should more clearly distinguish our setting from the action-conditioned world models commonly studied in RL. Therefore, we will revise the title to "*Structured 4D Latent Predictive Model for Robot Planning*", and update the abstract and introduction accordingly to make the positioning clearer.
>
> 2. (a) **Relation to other 3D planning and dynamics directions**. Thanks for providing the related 3D planning methods which we will discuss in the revised paper. The key difference is that our method learns **holistic scene-level 4D rollout** in 3D latent states, while methods such as Points2Plans and latent-space planning for multi-object manipulation focus on object-centric or relational planning abstractions. Compositional NeRF-based dynamics is an important 3D dynamics direction, but it is not directly an end-to-end robot planning pipeline under our evaluation setting. Among explicit 3D predictive models, GWM is the most directly related to our approach. However, runnable code for an end-to-end planning pipeline is not publicly available for GWM, which makes direct comparison not straightforward.
>     Following the reviewer's suggestion, we compared with 3D Diffuser Actor as a strong 3D policy baseline under the same ManiSkill3 StackCube setup. These two methods are comparable (91% vs. 90%) under the orginal settings, but our method is substantially more robust under stronger perturbations. We believe this supports the claim that predictive 3D rollout provides a meaningful robustness advantage beyond standard 3D policy learning.
>
>     |  |Original|Light|Noise (0.05)|Noise (0.08)|Noise (0.10)|BG Color|View (10$^\circ$)|
>     |---|---|---|---|---|---|---|---|
>     |Ours (CL)|91%|90%|86%|90%|86%|89%|88%|
>     |3D Diffuser Actor|90%|91%|85%|48%|7%|78%|88%|
>
>     (b) **Visual quality versus 3D consistency**. The reviewer is correct that the strongest empirical advantage of our method is on 3D consistency, rather than uniformly across standard 2D image metric. That is also the central property we aim to improve for manipulation planning. We will therefore narrow the wording of “visual quality” and state the claim more precisely: our method substantially improves 3D consistency and multiview coherence while remaining competitive on standard 2D image metrics, instead of claiming uniform better on all framewise appearance metrics.
>
> 3. **Role of the latent representation**. In our framework, the 3D latents serve as the **predictive model state** which encodes a compact 3D representation in a much lighter way than alternatives such as NeRF, 3D Gaussians, or dense point clouds. This compact 3D representation enables efficient conditioning and generation during future rollouts, while maintaining the ability to decode into explicit 3D representations such as 3D Gaussians and point clouds. For downstream control, we use the explicit point cloud as the **action interface** for the inverse dynamics, because it is an explicit 3D representation that is easier to translate into executable robot actions. Moreover, explicit point clouds also enable training-free motion-planning based inverse dynamics as discussed in the paper, which may help the framework generalize better.
>
> 4. **Scope and generality**. Thanks for this concern, and we agree that the current evidence is strongest in calibrated multiview settings, and the real-world validation is currently limited to one task. We therefore position this paper as the establishment and validation of the concept of 3D consistent predictive planning in robot planning, while broader extensions to more general cases is an important future works. We also believe the proposed framework has the potential to generalize more broadly, since in real-world settings, an embodied agent can actively explore the new viewpoints to obtain better 3D reconstructions.
>
> Overall, we appreciate the reviewer’s suggestions, especially for the positioning of the paper. These comments are very helpful, we will revise the paper accordingly to make the contribution and scope clearer in the final version.

---

> > ### Author Rebuttal · Reviewer_qiSB · 2026-04-06
> >
> > Thanks the authors for the rebuttal. My concerns are resolved and I raise my rating.

---

> > > ### Author Response · Authors · 2026-04-06
> > >
> > > Thank you for your positive follow up and confirming that our rebuttal has addressed the concerns. We sincerely appreciate your time, careful review, and your updated assessment.

---

### Decision · Program_Chairs · 2026-04-30

**Decision:**

Accept (regular)

**Comment:**

This paper initially received mixed ratings but the rebuttal addressed the main concerns effectively, and three of four reviewers confirmed their issues were resolved (with ratings of 1 weak accept, and 2 accept). While Reviewer uLdK expressed some persisting concerns about the data used to train baselines, the AC feels these were adequately addressed in the rebuttal, although the text should be improved to make these clearer. The AC agrees with the majority that the paper makes a solid contribution demonstrating clear 3D consistency advantages over video-based alternatives, and supports acceptance. As per the discussions with Reviewer  qiSB, the camera-ready should incorporate the promised retitling (away from "world model" framing) and positioning updates.